# Cord Blood Adipocytokines and Body Composition in Early Childhood: A Systematic Review and Meta-Analysis

**DOI:** 10.3390/ijerph18041897

**Published:** 2021-02-16

**Authors:** Christos Bagias, Nithya Sukumar, Yonas Weldeselassie, Oyinlola Oyebode, Ponnusamy Saravanan

**Affiliations:** 1Division of Health Sciences, Department of Population Evidence and Technologies, Warwick Medical School, University of Warwick, Coventry CV7 7HL, UK; chrisba1983@yahoo.gr (C.B.); N.Sukumar@warwick.ac.uk (N.S.); y.weldeselassie@warwick.ac.uk (Y.W.); o.r.o.oyebode@warwick.ac.uk (O.O.); 2Department of Endocrinology and Diabetes, University Hospital of Ioannina, 45500 Ioannina, Greece; 3Academic Department of Diabetes, Endocrinology and Metabolism, George Eliot Hospital NHS Trust, Nuneaton CV10 7DJ, UK

**Keywords:** cord blood leptin, cord blood adiponectin, adiposity

## Abstract

Childhood obesity is a growing epidemic. Early identification of high-risk groups will allow for the development of prevention strategies. Cord blood adipocytokines have been previously examined as biomarkers predicting future obesity. We conducted a systematic review looking at the association between cord blood leptin and adiponectin with adiposity up to 5 years of age. A literature review was performed between January 1994 and August 2020 using two bibliographic databases (Medline/Pubmed and EMBASE) and was registered on PROSPERO (CRD42017069024). Studies using skinfold thickness and direct methods of assessing body composition in full term neonates were considered. Partial correlation and multiple regression models were used to present the results. Meta-analysis was performed, were possible, using a random effects model. Cochran’s Q test was used to assess heterogeneity and I^2^ statistics to calculate the percentage of variation across studies. The potential for publication bias was assessed using funnel plots. Data from 22 studies were retrieved and reviewed by two independent reviewers. Cord blood leptin was positively associated with adiposity at birth (*r* = 0.487; 95% CI: 0.444, 0.531) but was inversely related to adiposity up to 3 years of age. The association was not sustained at 5 years. There was a weak positive association between adiponectin in cord blood and adiposity at birth (*r* = 0.201; 95% CI: 0.125, 0.277). No correlation was found between cord blood adiponectin in young children, but data were limited. This review supports that cord blood leptin and adiponectin are associated with adiposity at birth. The results of this study provide insight into the role of adipocytokines at birth on future metabolic health and their potential use as risk stratification tools.

## 1. Introduction

Childhood obesity has reached epidemic levels and constitutes one of the greatest public health challenges of the 21st century. Globally, the number of infants and young children up to the age 5 years affected by obesity are expected to increase to 70 million by 2025 [1]. This is associated with both short and long-term adverse outcomes. Children with obesity are likely to become adults with increased risk of diabetes, cardiovascular disease, osteoporosis, and cancer [2,3,4,5]. The origin of obesity is multifactorial, primarily caused by an imbalance between energy intake and expenditure. Sedentary lifestyle, poor diet, genetics, and epigenetics are central in the development of this [6]. There is increasing evidence that events occurring in early life, even before birth, can cause obesity. During fetal life the body goes through critical periods of development with lasting or life-long effects and, therefore, the intrauterine environment may program long-term trajectories of adiposity and metabolic health [7]. Identification of high-risk groups can help in developing preventative strategies for reducing childhood obesity and adverse cardiometabolic disorders.

Adipose tissue in infancy is essential in ensuring adequate energy supply to the brain in a period of nutritional disruption and acts as a thermo-insulator (white adipose tissue) and thermo-regulator (brown adipose tissue) [8]. While it has a protective role against infections, intensive care unit admissions, and mortality [9,10], the recent evidence shows that the strongest predictors of adult obesity are large for gestational age (LGA) and obesity in preschool children. However, the optimal range of adiposity in children that defines “beneficial versus metabolically harmful” levels of fat mass is yet to be identified [11]. To address this, researchers proposed body composition growth charts as opposed to birth weight charts to improve the predictive ability of future obesity [12]. Others tried to use cord blood adipocytokines such as leptin and adiponectin as objective markers to predict future obesity, in contrast to adiposity measurements, which are subjective and prone to measurement errors.

Leptin is predominantly produced by adipocytes [13]. Placenta, muscle, bone marrow, and stomach are other sites of leptin secretion [14,15,16,17]. Leptin regulates body weight through negative feedback between fat mass and hypothalamic centers of satiety [18]. In addition, it also controls basal metabolic rate. When it is bound to leptin receptors, expressed by many tissues, the energy expenditure at the cellular level increases [19]. Maternal leptin does not cross the placenta due to its high molecular weight [20]. In addition to fetal adipose tissue, the placenta is a major producer of leptin during fetal life and a significant contributor to cord blood leptin levels, despite 98% of its production being secreted into maternal circulation.

Adiponectin is almost exclusively produced by adipose tissue in adults [21]. It plays an important role in glucose and lipid metabolism, is inversely related to leptin, and low levels are observed in obesity [22], type 2 diabetes, and metabolic syndrome in adults [23]. Cord blood adiponectin is not related to maternal concentrations [24] and is not shown to be derived from placental or maternal tissue [25,26,27]. Apart from adipose tissue, adiponectin mRNA is also found in other tissues (muscle, kidney, skin) during fetal life [28]. Whether cord blood adiponectin simply mirrors fetal adiposity or also acts as a growth factor remains to be established.

Published studies reporting the association between cord blood adipocytokines and adiposity at birth show inconclusive results. Most were cross-sectional and in small cohorts [29,30,31,32,33,34,35,36,37,38,39,40,41,42]. Thus far, there have only been three moderately sized longitudinal [43,44,45] studies examining the link between cord blood levels and adiposity in early childhood. It is difficult to draw firm conclusions from these studies due to the different ethnic groups sampled and different methods in assessing the body composition. The purpose of this systematic review was to summarize these studies to improve the power and to shed light on the role of cord blood leptin and adiponectin on adiposity at birth and in early childhood.

## 2. Materials and Methods

### 2.1. Search Strategy and Study Selection

Observational, cross-sectional, and longitudinal studies were examined. Studies of the offspring of healthy pregnant women and those with obesity were included. For studies involving participants with diabetes (type 1 or 2, gestational diabetes), data obtained only from control groups were used, as offspring of mothers with diabetes have higher leptin concentrations for a given fat mass [46,47]. Studies of full term neonates of different growth patterns—appropriate for gestational age (AGA), small for gestational age (SGA), LGA—and ethnic origin were considered eligible. Studies examining neonates born preterm and/or with congenital abnormalities (chromosomal disease, respiratory distress syndrome, heart disease, and renal disease) were excluded. Randomized control trails were not included in the current systematic review. Due to the different criteria used to define SGA, LGA, and obesity across various populations, we accepted the authors’ definition.

Cord blood samples, measuring leptin and adiponectin, analyzed by enzyme-linked immunosorbent assay (ELISA) or radioimmunoassay (RIA) qualified for the meta-analysis. Leptin results were reported in ng/mL and adiponectin in μg/ml. We included studies that assessed neonatal and childhood adiposity by air displacement plethysmography (ADP), dual energy X-ray absorptiometry (DXA), magnetic resonance imaging (MRI), and anthropometric measurements (skinfold thickness). Studies using only ponderal index as a measure of adiposity were excluded.

The review was registered on PROSPERO (CRD42017069024). Meta-analysis of observational studies in epidemiology (MOOSE) guidelines were followed for the study [48]. Since the review involved synthesis of published data, National Health Service (NHS) Research Ethics Committee approval was not required.

A literature search was performed between January 1994 (discovery of leptin gene) and August 2020. Two bibliographic databases were used to conduct the searches: Medline/Pubmed (National Library of Medicine and National Institute of Health) and EMBASE (The Excerpta Medica Database). The following keywords and medical subject headings (MeSH) were used: “leptin”, “adiponectin”, “fetal blood”, “umbilical cord blood”, “adiposity”, “obesity”, “body composition”, “fetal growth”, and “anthropometry”. Search words were combined using Boolean operators (AND, OR). The search was limited further to those studies published in English and performed in humans from birth to 5 years of age. Reference lists from included studies were reviewed for further potentially eligible articles. If full-text online access was not available original journals were retrieved from the library services. Unpublished studies were not considered.

### 2.2. Data Extraction

Level one screening of search results (title and abstracts) was performed by two independent reviewers (CB and NS), using the inclusion and exclusion criteria provided. Level two screening of the full manuscripts was conducted by them independently (CB and NS) and any discrepancy was resolved by consulting a third reviewer (PS/OO). The PICO (Population, Intervention, Comparison, Outcome) framework was used to formulate the search strategy. For studies fulfilling the inclusion criteria, the independent reviewers extracted details onto standard data extraction templates. This included information about the type of study, location, and time of data collection, population, sample size, type of leptin and adiponectin assays, statistical methods, and technique to assess body composition. Any disagreement was resolved by discussion or by consulting a third reviewer.

### 2.3. Statistical Analysis

Partial correlation and multiple regression models were used to present the results. Studies were reported narratively, when correlation coefficient (*r*) was unable to be extracted. Meta-analysis of correlations was performed using a random effects model because of the varying population characteristics and sample sizes. Cochran’s Q test was used to assess heterogeneity and I^2^ statistics to calculate the percentage of variation across studies that was due to heterogeneity rather than chance. As different methods of assessing adiposity were used (DXA, ADP, skinfold thickness), the test of moderators was applied. We included “method” in the random effects model to examine the effect on heterogeneity. The results showed that there was no evidence for heterogeneity between the type of adiposity measures (QM test for moderators, *p* = 0.50). Hence in the final random effects model fitted we did not include the covariate “method”.

Forest plots were created for each outcome. The potential for publication bias was assessed using funnel plots when the requirement of ten or more studies per meta-analysis was met. Egger’s test to assess the funnel plots’ asymmetry was applied. All analyses were conducted using STATA version 14.

### 2.4. Assessment of Risk Bias

The assessment of methodological quality of the studies was done using checklist criteria. This systematic review included observational cohorts and cross-sectional studies, therefore, the quality assessment tool adopted from the National Institutes of Health/National Heart, Lung and Blood Institute was used (Appendix A). After answering a series of 14 questions, the quality of each study was reported as poor, fair, or good (Appendix A).

## 3. Results

After applying our search criteria, 169 studies were identified, which were reduced to 152 after removing duplicates. After title-abstract screening and full manuscript review, 22 studies [29,30,31,32,33,34,35,36,37,38,39,40,41,42,43,44,45,49,50,51,52,53,54] met all the inclusion and exclusion criteria. Figure 1 presents the selection of the studies included in the literature review.

### 3.1. Characteristics

Table 1 provides baseline characteristics of the studies included. Most of the studies examined White European neonates, while one study [49] examined an African American population. Three studies [32,33,37] included neonates of more than one ethnicity. In terms of maternal characteristics and risk factors, most of the studies included mothers without diabetes or other metabolic disorders. Two studies [30,34] reported the association between adipocytokines and neonatal adiposity based on intrauterine growth (SGA vs. AGA, AGA vs. LGA).

### 3.2. Studies Reporting Leptin

The correlation between leptin and adiposity was investigated in 17 studies [29,30,31,32,33,34,35,36,37,38,39,43,44,45,49,50,53]. Eleven [29,30,31,32,33,34,35,36,37,38,39] examined the relationship at birth using either Pearson correlation or multiple regression analysis. Eight [32,36,39,43,44,45,49,50] assessed the relationship between cord blood leptin and adiposity at different time points (3 weeks, 2 months, 2–5 years of age) using partial correlation or multiple regression models. Eleven studies [29,30,31,33,34,35,37,38,44,49,53] used radioimmunoassay (RIA) as the method of measuring cord blood leptin, whereas six [32,36,39,43,45,50] used enzyme-linked immunosorbent assay (ELISA). Body composition was assessed using DXA (*n* = 1) [31], ADP (*n* = 4) [33,36,37,49], MRI (*n* = 1) [50], or skinfold thickness (*n* = 12) [29,30,32,34,35,38,39,43,44,45,50,53]. Total body fat mass was derived by measuring skinfold thickness at two (triceps (TR) and subscapular (SS) [29,38,43,44,45]), three (TR + SS + quadriceps (QD) [35]), four (TR + SS + QD + suprailiac (SI) [30,32,39]), or six (TR + SS + QD + SI + biceps + gastrocnemius [34,50]) sites.

### 3.3. Studies Reporting Adiponectin

The correlation coefficient between adiponectin and adiposity at birth was examined in eight studies [30,32,33,34,40,41,42,51]. Martinez-Cordero et al. [30] described no association between cord blood adiponectin and adiposity. For the purpose of statistical analysis, this was considered as *r* = 0. Basu et al. [41] reported different correlation coefficients between male and female participants. Four studies [32,44,49,50] looked at the association between cord adiponectin and body composition at different age groups (1–3 months, 3–5 years of age). Adiponectin was measured using either RIA (*n* = 6) [30,33,34,44,49,52] or ELISA (*n* = 5) [32,40,41,42,51]. To calculate body composition, two studies used ADP [33,49], one MRI [51], and the remaining used skinfold thickness at different sites: 2 sites (*n* = 2) [44,51], 4 sites (*n* = 5) [30,32,40,42], or 6 sites (*n* = 1) [34].

Two studies [52,53] reporting the association between cord plasma adipocytokines and isolated skinfold measurements (not derived total adiposity) were excluded from further analysis.

### 3.4. Leptin and Neonatal Adiposity at Birth

All 11 studies [29,30,31,32,33,34,35,36,37,38,39] assessing 1653 pregnancies, revealed a moderate, positive correlation between cord blood leptin and neonatal adiposity (random effect model; *r* = 0.487; 95% CI: 0.444, 0.531; Figure 2). Eight studies reported mean levels of cord blood leptin [32,33,34,37,43,44,45,50]. Applying the continuous random effects model revealed a pooled mean value of 9.1 ng/mL (95% CI: 8.27, 9.95; *p* < 0.001). However, a high level of heterogeneity was present (I^2^ = 85.76%, *p* < 0.001).

### 3.5. Leptin and Adiposity in Early Childhood

Eight studies reported the correlation between cord blood leptin and adiposity in early childhood at varying times [32,36,39,43,44,45,49,50]. Meta-analysis was not possible due to the varying times of follow-up, nature of reporting of adiposity (absolute vs. change in skinfold/fat mass), and the adjustments for differing maternal characteristics in the studies. The study results are summarized in Table 2.

In a small study of African American babies (*n* = 36), no correlation between cord leptin and fat mass at 2 weeks and 3 months of life was observed [49]. However, in 221 babies studied by Chaoimh et al., cord blood leptin was inversely associated with fat mass index at 2 months of age (β: −0.021, 95% CI: −0.034, −0.007, *p* = 0.003) [36]. Similarly, two other studies (*n* = 188 and 508) [39,45] demonstrated an inverse correlation between cord leptin and fat mass at 2 years and sum of skinfolds (SSF) thickness at 3 years of age after adjusting for maternal and offspring characteristics. In concordance, Mantzoros et al. [44] (*n* = 588) reported an inverse relationship with fat mass at 3 years of age, although not statistically significant (β: −0.24, 95% CI: −0.88, 0.41; *p* = 0.48). However, it is likely that this study and the one by Boeke et al. [45] are from the same cohort (Project Viva) and had opposite findings presumably due to different skin fold thicknesses and follow-up numbers used. Meyer et al. [50], following a cohort of 89 offspring, showed that cord leptin was negatively associated with total fat mass (calculated using skinfold thicknesses) at 3 and 5 years of age (3 years: β: −0.02, 95% CI: −0.04, −0.00, *p* = 0.03; 5 years: β: −0.03, 95% CI: −0.06, −0.00, *p* = 0.03). The association was not observed when fat mass was assessed by MRI at 5 years in a subgroup of 33 children (visceral adipose tissue: β: 0.26, 95% CI: −1.71, 2.23, *p* = 0.78; subcutaneous adipose tissue: β: −0.13, 95% CI: −9.20, 8.94, *p* = 0.97). Two other studies revealed no associations with fat mass at 4 (SSF derived) and 7 years (DXA derived) of age (Table 2) [43,45].

### 3.6. Adiponectin and Neonatal Adiposity at Birth

Pooled effect of the nine studies [30,32,33,34,40,41,42,51] (*n* = 869) revealed a weak positive correlation between cord blood adiponectin and neonatal fat mass (random effects model; *r* = 0.201; 95% CI: 0.125, 0.277; Figure 3). Four studies (*n* = 450) reported mean adiponectin levels with a pooled mean adiponectin at birth of 25.6 μg/mL (95% CI: 16.5, 24.76). Again, a high level of heterogeneity was observed (I2 = 99.77%, *p* < 0.001) [33,34,51,52].

### 3.7. Adiponectin and Adiposity in Early Childhood

The studies included presented mixed results regarding the relationship of cord blood adiponectin with weight gain and adiposity in different age groups. Meta-analysis was not possible, and the study results are summarized in Table 3. Teague et al. [32] (*n* = 52), showed a positive correlation of high molecular weight adiponectin (HMWA) with both weight gain (*r* = 0.40, *p* = 0.003) and adiposity (*r* = 0.32, *p* = 0.02) at 1 month of age. In contrast, cord adiponectin negatively predicted adiposity (*r* = −0.38, *p* < 0.05) in 36 African American infants at 3 months of age [49].

Mantzoros et al. [44] (*n* = 588) showed that cord blood adiponectin was not associated with total fat mass (SS + TR thickness) at 6 months (β: 0.42, 95% CI: −0.11, 0.95, *p* = 0.12) but positively with central adiposity (SS/TR thickness) at 3 years of age (β: 2.01, 95% CI: 0.09, 3.93, *p* = 0.04). Similarly, Meyer et al. [51], examining both total and HMWA, noted a positive association with % fat mass at 3 years of age (β: 0.04, 95% CI: 0.00, 0.08, *p* = 0.04) but not at 5 years (β: 0.02, 95% CI: −0.04, 0.08, *p* = 0.5; Table 3).

### 3.8. Publication Bias

Publication bias was assessed via a funnel plot only for the leptin group, as the minimum requirement for ten studies per meta-analysis was not met in the adiponectin group. Application of the Egger’s test did not reveal any asymmetry of the funnel plot, indicating no evidence of publication bias (Figure 4).

## 4. Discussion

We observed three key findings in this systematic review and meta-analysis. First, cord leptin and adiponectin levels were positively associated with adiposity at birth. The association was stronger for leptin than for adiponectin. Second, the association between leptin with adiposity was inverse in early childhood. Third, adiponectin appears to be positively associated with adiposity in early childhood, although the data for this are limited.

### 4.1. Leptin and Adiposity

Maternal leptin cannot cross the placenta due to its high molecular weight, and it is not related to cord blood levels [20]. Cord leptin is primarily derived from fetal adipose tissue although it can also come from the placenta. While the majority of placental leptin drains into the maternal circulation, small amounts enter the fetal circulation [54]. Leptin levels are higher in the umbilical artery compared to that in the umbilical vein [55]. Fetal leptin is detectable at as early as 18 weeks of gestation, with levels rising as pregnancy progresses, in concordance with fetal fat accumulation [56]. In addition, the presence of leptin mRNA and leptin receptors in various fetal tissues implies its role as a growth factor during intrauterine development. All these findings support that cord leptin is a good marker of adiposity at birth.

Our analysis revealed that cord leptin levels are inversely related to adiposity at 3 years of age [39,44,45,50], but this was not seen at 4 and 7 years [43,45]. Chaoimh et al. [36] noted that cord leptin is inversely associated with adiposity gain from birth to 2 months of age, by an objective adiposity measure, ADP. In our study, pooled mean leptin level at birth was 9.1 ng/mL (*n* = 2242). Although, there is no reference range for cord blood leptin levels, Karakosta et al. [57] observed a mean leptin value of 7.7 ng/mL in a cohort of 398 healthy, full term neonates born in Greece. Taken together, it appears that children born with higher leptin levels, develop a “compensatory behavior” driven by the anorexigenic effect of leptin in early postnatal life, which lasts up to 3 years of age. In order to maintain a positive energy balance and enhance vital organ development, leptin’s full metabolic effect is not exerted before the second postnatal week [58]. The above observation could potentially explain the positive association between cord leptin and adiposity during the first weeks of life, as described by Schneider et al. [49] and Teague et al. [32].

The above conclusion is also supported by evidence from animal studies. Leptin plays an important role in brain development. The human hypothalamus develops predominantly during the prenatal period [59]. The arcuate nucleus (ARC), the major site for energy regulation, develops after 34 weeks of gestation, but further changes take place in the early postnatal life. Independent of fat mass accretion, a leptin surge happens in the immediate postnatal period, which is critical for the development of projections from the ARC to paraventricular hypothalamic nuclei [60]. Ob/ob mice have impaired projections, an effect which can be reversed by early life leptin administration, whereas leptin administration in adulthood has no effect, implying a tight window for leptin’s neurotrophic action [61].

Thus, raised leptin levels at birth may adversely program the hypothalamus (via an impaired leptin surge), with effects becoming evident after the third year of life. The initial inverse correlation between cord leptin and adiposity may be due to the anorexigenic effect of leptin, followed by leptin resistance, resulting in hyperphagia and increased adiposity. In the present systematic review, Meyer et al. [50] and Boeke et al. [45] used direct measures of adiposity (MRI, DXA) to assess the association between cord leptin and adiposity beyond the third year of life. Both studies demonstrated a trend for cord leptin to inversely predict adiposity at 3 years of life, which was converted to a positive association by the ages of 5 and 7. Boeke et al. [45] also highlighted that serum leptin at 3 years of age positively predicts adiposity at 7 years of age, results consistent with leptin resistance.

### 4.2. Adiponectin and Adiposity

Adiponectin levels increase by 20-fold from midgestation to term [62]. Cord levels are not related to maternal adiponectin [24,63] and placental production is not yet confirmed [25,64]. Sivan et al. [27] showed that cord blood adiponectin levels were similar to those four days postpartum, confirming independent fetal production. The inverse correlation between adipose tissue and adiponectin observed in the adult population is not present during the early stages of life. Evidence suggests that the shift from positive to negative correlation between adiponectin and fat mass occurs around school-entry age [65]. Multiple sites of adiponectin expression other than adipose tissue during fetal life [28], such as brown adipose tissue [66], the subcutaneous-to-visceral fat ratio of neonates [67], and the inhibition of adiponectin expression from inflammatory cytokines produced by hypertrophic adult adipocytes [68] could potentially explain the different metabolic profile between neonatal and adult life.

The predictive capacity of cord adiponectin for the development of future adiposity remains unclear. Small sample size, different methods to measure adiponectin (ELISA, RIA), and different multimeric forms examined may contribute to the inconsistent results. Simpson et al. [69] recently reviewed the association between adiponectin and adiposity at 9 and 17 years of age. Results showed no correlation with adiposity at 9 years and a positive correlation at 17 years with a very small effect size (β: 0.02, 95% CI: 0.00, 0.03, *p* < 0.05). Major limitations of the study were the big loss to follow-up and the small percentage of children with obesity (sample not representative of most populations), which could have potentially attenuated any associations.

### 4.3. Strengths and Weakness

To our knowledge, our systematic review and meta-analysis is the first to assess the correlation of cord blood adipocytokines with adiposity at different time points. However, while it had reasonable sample size to assess the independent associations of leptin and adiponectin with adiposity at birth, it did not have adequate sample size for early childhood. In addition, as this review included observational cohorts and cross sectional studies, these associations do not prove causality. We also acknowledge that in the majority of the studies fat mass was derived from the sum of skinfolds and not by direct measures of adiposity, which would have minimized intra- and inter-subject variability. Studies investigating the relationship between ELISA- and RIA-based methods are rare, therefore, the assays used to measure adipocytokines may have contributed to the variability of the results. Finally, studies adjusted their results for different confounding variables. Maternal body mass index (BMI) was considered in all studies, whereas only half of the included studies [29,36,37,40,43,44,45,49,53] controlled for smoking during pregnancy, a factor inversely related to cord blood adipocytokines as well as birth size.

## 5. Conclusions

The present systematic review and meta-analysis reveals positive associations between cord blood leptin, adiponectin, and body fat at birth. However, cord blood leptin inversely predicts adiposity in early childhood, around 3 years of age, suggesting the early development of leptin resistance, although data for this are limited. In order to understand the origins of obesity and metabolic diseases, we need to investigate the trajectory of body composition (not just by weight) in early childhood and its relationship with key metabolic factors. Body weight, BMI, and skinfold measurements do not reflect fat mass (FM) precisely and are prone to inter- and intra-observer variation. However, objective measures of adiposity are cumbersome and not widely available. Biochemical markers in cord blood such as leptin are easy to measure and are potentially useful for risk stratification of children at birth for obesity and other related cardiometabolic disorders in later life. This could help in targeted, individualized prevention strategies to be implemented from birth/early childhood. Additional studies on the long-term effects of leptin and adiponectin at birth are required to confirm our findings on the predictive value on future adverse cardiometabolic risk and to understand the potential underlying mechanisms.

## Figures and Tables

**Figure 1 ijerph-18-01897-f001:**
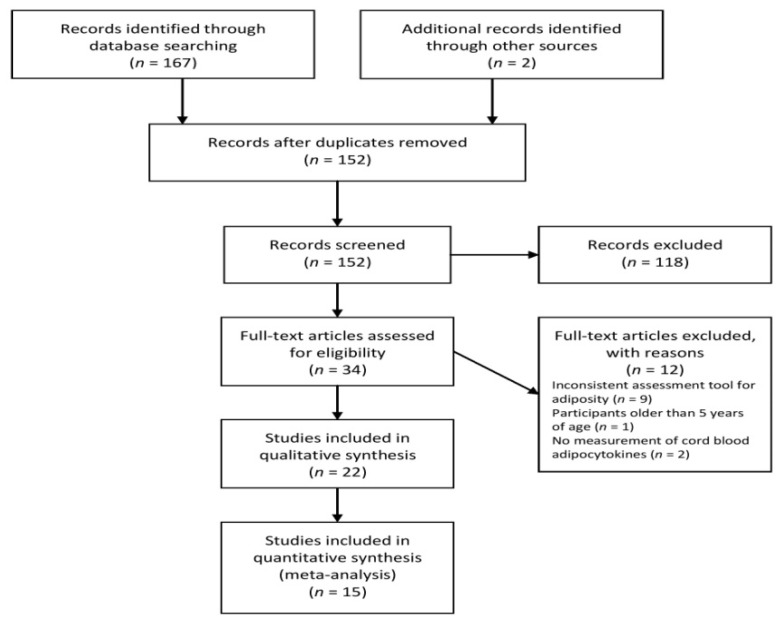
Flow diagram of the study.

**Figure 2 ijerph-18-01897-f002:**
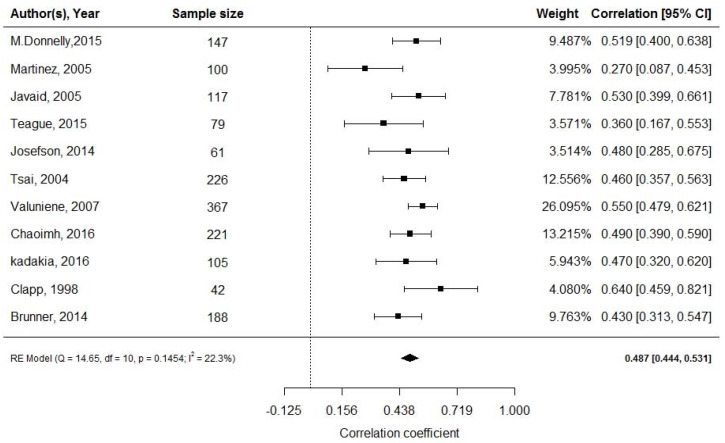
Cord blood leptin and fat mass at birth; df: degree of freedom, RE: random effects model.

**Figure 3 ijerph-18-01897-f003:**
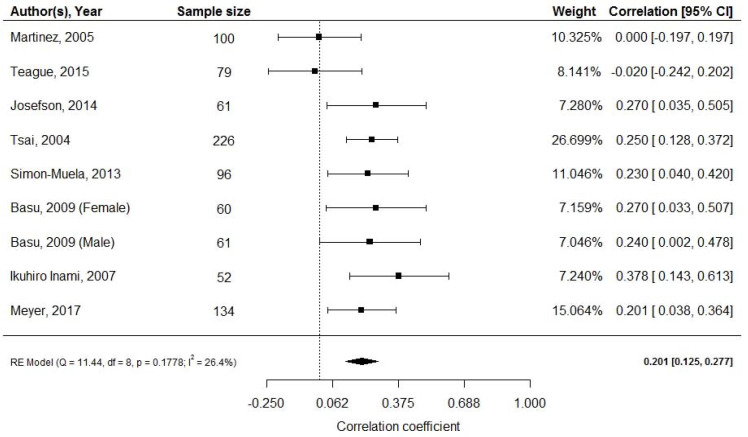
Cord blood adiponectin and fat mass at birth; df: degree of freedom, RE: random effects model.

**Figure 4 ijerph-18-01897-f004:**
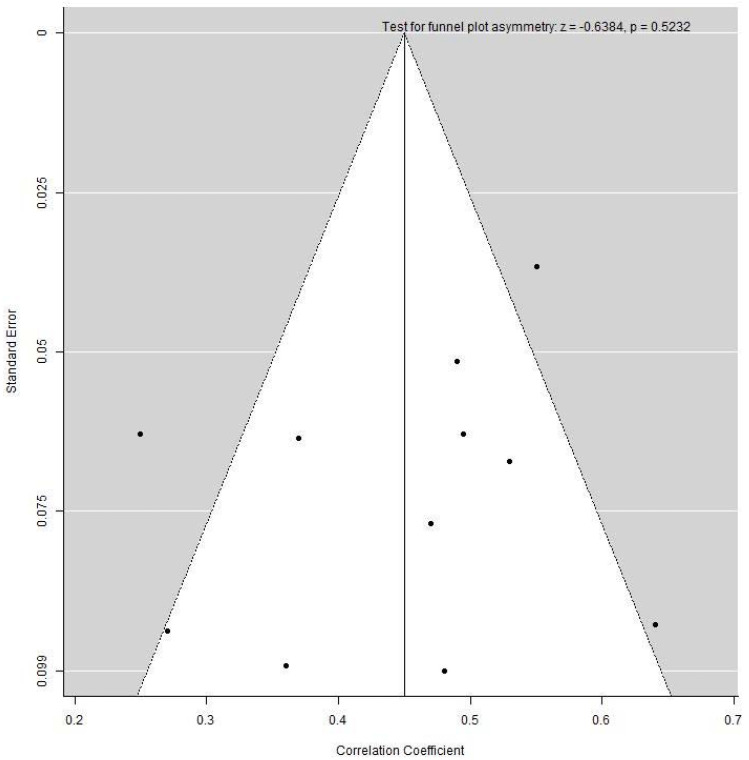
Funnel plot examining heterogeneity in the cord blood leptin and fat mass at birth group.

**Table 1 ijerph-18-01897-t001:** Characteristics of the studies included.

Study	Country (Sample Size)	Year of Data Collection	Follow-Up Duration	Assay Method		Adiposity Method	Limitations	Author’s Conclusion
				Leptin	Adiponectin			
Meyer, 2018 [50]	Germany (*n* = 89)	NR	5 years	ELISA	-	Skinfold, MRI	Drop out, maternal BMI	L does not predict adiposity
Meyer, 2017 [51]	Germany (*n* = 90)	NR	5 years	-	ELISA	Skinfold, MRI	Drop out, maternal BMI	Ad does not predict adiposity
Schneider, 2017 [49]	USA (*n* = 36)	NR	3 months	RIA	RIA	ADP	Sample size	Ad but not L predicts adiposity
Kadakia, 2016 [37]	USA (*n* = 105)	2011–2014	At birth	RIA	-	ADP	Sample size	Early life L resistance
Karakosta, 2016 [43]	Greece (*n* = 578)	NR	4 years	ELISA	-	Skinfold	NR	L does not predict adiposity
Chaoimh, 2016 [36]	Ireland (*n* = 221)	2008–2011	2 months	ELISA	-	ADP	Sample size, maternal BMI, infant feeding not quantified	L predicts adiposity, longitudinal data required
Donnelly, 2015 [29]	Ireland (*n* = 147)	2007–2011	At birth	RIA	-	Skinfold	Selection bias	L predicts adiposity
Teague, 2015 [32]	USA (*n* = 124)	2010–2013	1 month	ELISA	ELISA	Skinfold, DXA	Drop out	L and Ad predict adiposity
Josefson, 2014 [33]	USA (*n* = 61)	NR	At birth	RIA	RIA	ADP	Sample size	L and Ad correlate with FM
Brunner, 2014 [39]	Germany (*n* = 188)	NR	2 years	ELISA	-	Skinfold	Sample size	L predicts adiposity
Boeke, 2013 [45]	USA (*n* = 508)	1999–2002	7 years	ELISA	-	Skinfold	Circadian L variation	L predicts adiposity
Simon-Muela, 2013 [40]	Spain (*n* = 96)	NR	At birth	ELISA	ELISA	Skinfold	NR	Ad action shows sex dimorphism
Basu, 2009 [41]	USA (*n* = 121)	NR	At birth	-	ELISA	Skinfold	Ad assay	Ad correlates with FM, sex dimorphism
Mantzoros, 2009 [44]	USA (*n* = 588)	1999–2002	3 years	RIA	RIA	Skinfold	Socioeconomic status	Ad but not L predicts adiposity
Inami, 2007 [42]	Japan (*n* = 52)	2004–2005	1 month	-	ELISA	Skinfold	NR	Ad correlates with FM
Valuniene, 2007 [35]	Lithuania (*n* = 367)	1998–2000	At birth	RIA	-	Skinfold	NR	L correlates with FM
Martinez, 2005 [30]	Mexico (*n* = 100)	NR	At birth	RIA	RIA	Skinfold	Sample size	L but not Ad correlates with FM
Javaid, 2005 [31]	England (*n* = 117)	NR	At birth	RIA	-	DXA	Sample size	L correlates with bone and FM
Tsai, 2004 [34]	Taiwan (*n* = 226)	2001–2002	At birth	RIA	RIA	Skinfold	NR	L and Ad correlate with FM
Lindsay, 2003 [52]	Scotland (*n* = 73)	1999–2001	At birth	-	RIA	Skinfold	NR	Ad not associated with skinfold
Geary, 1999 [53]	England (*n* = 39)	1996–1997	At birth	RIA	-	Skinfold	NR	L correlates with FM
Clapp, 1998 [38]	USA (*n* = 42)	NR	At birth	RIA	-	Skinfold	NR	L correlates with FM

Ad: adiponectin, ADP: air displacement plethysmography, BMI: body mass index, DXA: dual energy absorptiometry, ELISA: enzyme-linked immunosorbent assay, FM: fat mass, L: leptin, MRI: magnetic resonance imaging, NR: not reported, RIA: radioimmunoassay.

**Table 2 ijerph-18-01897-t002:** Cord blood leptin and adiposity in early childhood.

Study (Sample size)	Adiposity Measure Studied	Adjustments	Results	*p*-Value
Schneider, 2017 [49] (*n* = 36)	FM (g) at 2 weeks and conditional change from 2 weeks to 3 months	2 weeks: gestational age, age at measurement, FFM 3 months: above + 2 weeks measurement and time between measurements	2 weeks: *r* = 0.27 3 months: *r* = −0.19	*p* > 0.05
Chaoimh, 2016 [36] (*n* = 221)	Conditional change to FMI (kg/m^2^) from birth to 2 months	Maternal age-education-smoking, maternal BMI at 15 weeks gestation, family income, sex, gestational age, breast feeding	Β-Coef (95% CI): −0.021 (−0.034, −0.007)	*p* = 0.003
Brunner, 2014 [39] (*n* = 90) ^a^	FM (g) at 2 years	Maternal BMI, gestational weight gain, pregnancy duration, sex, breast feeding	Β-Coef (95% CI): −14.86 (−29.49, −0.23)	*p* = 0.04
Boeke, 2013 [45] (*n* = 508) ^b^	SSF (mm) at 3 years SSF (mm) and DXA derived FM (kg) at 7 years	Maternal age, weight gain, income, education, smoking, sex, ethnicity, breast feeding	3 years Β-Coef (95% CI0: −1.4 (−2.7, −0.1) 7 years Β-Coef (95% CI): 1.1 (−1.5, 2.1) for SSF 0.3 (−0.7, 1.3) for DXA	*p* < 0.05 *p* > 0.05
Mantzoros, 2009 [44] (*n* = 588) ^b^	SS + TR (mm) and SS/TR (mm) at 3 years of age (regression per 10 ng/mL of leptin)	Maternal education, pre-pregnancy BMI, weight gain, gestational age, paternal BMI, sex, ethnicity, breast feeding	Β-Coef (95% CI): −0.24 (−0.88, 0.41) for SS + TR −0.22 (−2.61, 2.17) for SS/TR	*p* = 0.48 *p* = 0.86
Meyer, 2018 [50] (*n* = 89) ^a^	% FM at 3 and 5 years VAT (cm^3^) and SAT (cm^3^) at 5 years	Maternal BMI, gestational weight gain, pregnancy duration, sex, breast feeding	Β-Coef (95% CI) % FM at 3: −0.06 (−0.13, 0.01) % FM at 5: −0.09 (0.17, 0.00) VAT: 0.26 (−1.71, 2.23) SAT: −0.13 (−9.20, 8.94)	*p* = 0.07 *p* = 0.04 *p* = 0.78 *p* = 0.97
Karakosta, 2016 [43] (*n* = 578)	SSF (mm) at 4 years of age	Sex, birthweight, maternal age and education, parity, pre-pregnancy BMI, breast feeding duration	Β-Coef (95% CI): 0.2 (−1.4, 1.7)	*p* > 0.05
Teague, 2015 [32] (*n* = 52)	% FM at 1month	Diabetic status, sex, age in days	*r* = 0.19	*p* = 0.19

BMI: body mass index, DXA: dual energy absorptiometry, FM: fat mass, FFM: fat free mass, FMI: fat mass index, SSF: sum of skinfolds, SS: subscapular, TR: triceps, VAT: visceral adipose tissue, SAT: subcutaneous adipose tissue. ^a^/^b^ indicates that these two studies are possibly from the same cohort with different durations of follow-up.

**Table 3 ijerph-18-01897-t003:** Cord blood adiponectin and adiposity in early childhood.

Study (Sample Size)	Adiposity Measure Studied	Adjustments	Results	*p*-Value
Teague, 2015 [32] (*n* = 52)	% FM at 1 month	Diabetic status, sex, age in days	*r* = 0.32	*p* = 0.02
Schneider, 2017 [49] (*n* = 36)	FM (g) at 2 weeks and conditional change from 2 weeks to 3 months	2 weeks: gestational age, age at measurement, FFM 3 months: above + 2 weeks measurement and time between measurements	2 weeks: *r* = 0.45 3 months: *r* = −0.38	*p* < 0.001 *p* < 0.05
Mantzoros, 2009 [44] (*n* = 588)	SS + TR (mm) and SS/TR (mm) at 3 years of age (regression per 10 μg/mL of adiponectin)	Maternal education, pre-pregnancy BMI, weight gain, gestational age, paternal BMI, sex, ethnicity, breast feeding	Β-Coef (95% CI): 0.42 (−0.11, 0.95) for SS + TR 2.01 (0.09, 3.93) for SS/TR	*p* = 0.12 *p* = 0.04
Meyer, 2017 [51] (*n* = 90)	% FM at 3 and 5 years VAT (cm^3^) and SAT(cm^3^) at 5 years	Maternal BMI, gestational weight gain, pregnancy duration, sex, breast feeding	Β-Coef (95% CI) % FM at 3: 0.21 (0.06, 0.35_ % FM at 5: 0.08 (−0.10, 0.27) VAT: 1.57 (−2.20, 5.34) SAT: 7.22 (−10.17, 24.62)	*p* < 0.05 *p* = 0.36 *p* = 0.39 *p* = 0.40

BMI: body mass index, FM: fat mass, FFM: fat free mass, SS: subscapular, TR: triceps, VAT: visceral adipose tissue, SAT: subcutaneous adipose tissue.

## Data Availability

The data presented in this study are available on request from the corresponding author. The data are not publicly available due to the standard operating procedures of the corresponding author’s institutional policy for systematic review and meta-analysis.

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
