# Peer review of "Cord Blood Adipocytokines and Body Composition in Early Childhood: A Systematic Review and Meta-Analysis"

_ijerph, 2021, doi:10.3390/ijerph18041897_

Round 1

Reviewer 1 Report

In reviewers opinion, it is a very interesting  and informative study, the meta-analysis

that analysing the association between cord blood leptin and adiponectin levels with children’s obesity in early childhood. Authors selected only healthy pregnant women, eliminating patients with diabetes.

Performed analysis reveals positive associations between cord blood leptin, adiponectin and neonatal body fat mass at birth. Authors observed three key findings in this systematic review. First, cord leptin and adiponectin are positively associated with adiposity at birth. The association was stronger for leptin than adiponectin. The very interesting finding is that the association between leptin with adiposity was inverse in early childhood and that adiponectin appears to be positively associated with adiposity in early childhood. Authors try to find the explanation for this relatively surprising findings.

The methodology of this review is proper and adequate for performed analysis.

I recommend this review for publication; the only remark is to remove in

vers 329 in the word tissue two digits 66.

Reviewer 2 Report

The manuscript entitled “Cord blood adipocytokines and body composition in early childhood: A systematic review” presents interesting issue, but some areas must be corrected.

Major:

Authors should stay consistent in their study and all the statements should be justified – they either prepared meta-analysis or not. As they did not prepare meta-analysis, as it was not possible, they can not suggest in any other section, that they did it (e.g. Conclusions: “The present meta-analysis reveals positive associations”), as it may be perceives as ethical problem.

General:

Authors should prepare their systematic review as adhering to the Preferred Reporting Items for Systematic Review and Meta-analysis (PRISMA), which are very specific and should be rigorously followed, to enable readers easier getting familiar with the study (to find specific information in each section). Authors should get familiar with PRISMA checklist (http://prisma-statement.org/prismastatement/Checklist.aspx) and they should correct their manuscript to be prepared according to the checklist. E.g., the Abstract should include: background, objectives, data sources, study eligibility criteria, participants, and interventions, study appraisal and synthesis methods, results, limitations, conclusions and implications of key findings, systematic review registration number, while a number of elements is not presented in the Abstract of the submitted manuscript. However, Authors should correct the whole study (not only the Abstract Section).

English language should be polished in whole manuscript (e.g. why “Public Health” is wrote with capital letters?)

Abstract:

For the systematic review readers this section is crucial and it requires corrections to provide all the necessary information (see above).

Introduction:

“Published studies examining the association between cord blood adipocytokines and adiposity at birth show mixed results” – such information should not be presented in this section, as this association is to be studied in this paper

Lines 80-82- hypothesis should not be presented in this section, but rather Materials and Methods Section

Materials and Methods:

Line 107-108 – this information suggest that lack of agreement of ethical committee was a reason to reject the study – if so, it should be clearly stated

Results and Discussion:

Figure 1 – should be polished, e.g. information about keywords should be removed from the figure. Moreover, PRISMA flowchart is for readers easier to follow (http://prisma-statement.org/PRISMAStatement/FlowDiagram)

The major part of the manuscript is large table (Table 1 – pages 6-7), which is does not present the essential information. Authors should divide this large table into several smaller ones and include to them also other important information form the included studies (e.g. time when the study was conducted, detailed observations, conclusions formulated by authors of the included studies).

Authors should either change the section from “Results” to “Results and Discussion”, or include additional section presenting discussion (preferably – such structure is for readers easier to follow)

Authors should in their discussion: (1) formulate clearly implications of the results of their study, (2) formulate clearly the future areas which should be studied.

Authors Contribution:

It should be clearly stated that all Authors participated in preparing manuscript.

Reviewer 3 Report

Congratulations to the reviewers for their approach to the article and for their work.

The article is well thought out and interesting but has some important areas for improvement, although they can be solved by the authors.

In the journal's instructions to the author, it states "Reviews: These provide concise and precise updates on the latest progress made in a given area of research. Systematic reviews should follow the PRISMA guidelines. "

The title should include the word meta-analysis.

The introduction is well written, clarifies the subject matter and the objective of the study.

The subject of the article is interesting. Although it is curious that being a current topic in the review, 22 articles have been selected, up to August 2020, the same as the authors selected in their February 2019 communication where they put the search date as November 2018. (https://www.morressier.com/article/cord-blood-adipocytokines-body-composition-5-years-age-systematic-review/5c3379afe668b90015afa000)

The article moves from point 3.11 (Line 342) to 5. Conclusions (line 357). The results are presented divided into a large number of conceptual aspects, but no discussion is offered that seeks any explanation of the data obtained from the analysis of the results.

Line 105 a mistake in reference 48 , no brackets.

Some of the references are not dated (e.g. 29,30,31,32,35,36....), which makes it difficult to assess the timeliness of the references.

Round 2

Reviewer 2 Report

The manuscript entitled “Cord blood adipocytokines and body composition in early childhood: A systematic review” presents interesting issue and was adequately corrected by Authors.

Reviewer 3 Report

Congratulations to the authors for the improvements in the article.